# Bacillus Subtilis (BG01-4^TM^) Improves Self-Reported Symptoms for Constipation, Indigestion, and Dyspepsia: A Phase 1/2A Randomized Controlled Trial

**DOI:** 10.3390/nu15214490

**Published:** 2023-10-24

**Authors:** Craig Patch, Alan J. Pearce, Mek Cheng, Ray Boyapati, J. Thomas Brenna

**Affiliations:** 1School Allied Health, Human Services and Sport, Melbourne 3086, Australia; 2Adepa Lifesciences, Melbourne 3000, Australia; 3Department of Gastroenterology, Monash Health, Melbourne 3977, Australia; 4Faculty of Medicine, Nursing and Health Science, Monash University, Melbourne 3800, Australia; 5Division of Nutritional Sciences, Cornell University, Ithaca, NY 14853, USA; 6Dell Paediatric Research Institute, Department of Paediatrics, of Nutrition, and of Chemistry, University of Texas at Austin, Austin, TX 78712, USA

**Keywords:** probiotics, *Bacillus subtilis*, functional gastrointestinal disorders, Gastrointestinal Symptom Rating Scale

## Abstract

Background: Functional gastrointestinal disorders (FGIDs) are common, difficult-to-manage conditions. Probiotics are emerging as a dietary component that influence gastrointestinal (GI) health. We conducted a double-blinded randomised controlled trial of a proprietary strain of deactivated *Bacillus subtilis* (BG01-4™) high in branched-chain fatty acids (BCFA) to treat self-reported FGID. Methods: Participants (n = 67) completed a four-week intervention of BG01-4™ (n = 34) or placebo (n = 33). The Gastrointestinal Symptom Rating Scale (GSRS) served as the outcome measure, collected prior to, at two weeks, and at four weeks after completion of the intervention. Results: At four weeks, one of three primary outcomes, constipation in the experimental group, was improved by 33% compared to placebo (15%); both other primary outcomes, Total GSRS and diarrhoea, were significantly improved in both the experimental and placebo groups (32%/26% and 20%/22%, respectively). The pre-planned secondary outcome, indigestion, was improved at four weeks (32%) but compared to the placebo (21%) was not significant (*p* = 0.079). Exploratory analysis, however, revealed that clusters for constipation (18% improvement, *p* < 0.001), indigestion (11% improvement, *p* = 0.04), and dyspepsia (10% improvement, *p* = 0.04) were significantly improved in the intervention group compared to the placebo. Conclusions: These initial findings suggest that in people with self-reported FGID, BG01-4™ improves specific symptoms of constipation and related GI dysfunction. Longer-term confirmatory studies for this intervention are warranted. Trial registration: This study was registered prospectively (25 October 2021) at the Australian New Zealand Clinical Trials Registry (ACTRN12621001441808p).

## 1. Introduction

Functional gastrointestinal disorders (FGIDs) are a heterogeneous group of chronic conditions characterised by persistent and recurrent gastrointestinal symptoms without any apparent structural or biochemical abnormalities [1,2,3]. Affecting between 25 and 40% of the global population, the most common of these conditions include functional bowel disorders and irritable bowel syndrome, as well as functional constipation, functional diarrhoea, and functional bloating [2]. Constipation or diarrhoea can be a burdensome FGID issue that occurs in many individuals and whose treatment remains challenging. Similarly, bloating is a subjective abdominal discomfort that is associated with abdominal inflation due to the accumulation of excessive gas (flatulence) and may not necessarily be accompanied by abdominal distension [4,5,6]. Whereas constipation and diarrhoea can be classified using the Bristol Stool Form Scale, bloating symptoms are generally subjective in nature, and therefore, assigning an objective score is fraught with inter- and intra-subject variations. However, the severity of bloating-related abdominal discomfort experienced by individuals can range from mild to severe, and the discomfort can negatively impact health and mental health [2]. These conditions carry significant morbidity, often leading to missed work/school and social isolation, and are commonly associated with psychological comorbidity. FGID presents a significant health-care burden, with over two-thirds of people with FGID having sought medical advice from a doctor in the preceding 12 months, 40% having been prescribed regular medication, and around a third undergoing potentially unnecessary abdominal surgery for their symptoms [7,8].

The pathophysiology and mechanisms underlying FGID are complex and not well understood. Consequently, although FGID is very common, it is a difficult-to-manage problem. Several mechanisms have been proposed, including abnormal gastrointestinal motility, visceral hypersensitivity, psychosocial disturbances, and, more recently, low-grade intestinal inflammation, mucosal immune dysregulation, and altered gut microbiota. Current therapeutic strategies are directed by symptomatology, but established treatments such as antibiotics and anti-depressants tend to be inconsistently successful or potentially contraindicated for long-term use [4,5,9]. Dietary restriction (e.g., low-FODMAPs) is a well-established therapy for some types of FGID, although there is no single diet that will help resolve symptoms for all patients. A recent population-based cross-sectional study by Stocks et al. [10] suggested that one-third of adults with a gastrointestinal condition were restricting their diet, which affected their mental health and overall quality of life.

Branched-chain fatty acids (BCFAs) were first recognised as nutrients important to gastrointestinal (GI) health in the context of vernix caseosa, the white waxy substance that develops on late-term human foetuses. Vernix caseosa sloughs into amniotic fluid as particles that are swallowed by the foetus throughout the last trimester [11]. Vernix is rich in BCFAs, which constitute about 30% of all fatty acids. BCFAs are rapidly taken up and incorporated into enterocyte membranes, where they moderate the inflammatory response [12,13,14].

Many microorganisms use BCFAs in their membranes to modulate biophysical properties [15]. Prominent among these is the Gram-positive *Bacillus* genus, which synthesises BCFAs de novo [16]. *Bacillus subtilis* (*B. subtilis*), the most well-studied Gram-positive microorganism, generally has an excess of 50% of BCFAs in its membrane lipids [17]. Populations of *Bacillus* used as probiotics [18,19] are safe in quantities of up to 10 × 10^9^ colony-forming units per day [20]. Specifically, certain species of Bacillus (e.g., *B. subtilis*) that are present in baked and fermented food products have a long and safe history of use in humans [5,21,22].

*B. subtilis* produces spores that remain viable in a wide temperature and pH range, making it a suitable probiotic supplement for improving gut health. Further, *B. subtilis* has no known contraindications. In concentrated form, safety studies have reported that doses of 2 × 10^9^
*B. subtilis* spores per day administered for a total 40 days (4 consumption periods of 10 days separated by 18 day washouts) were safe and well-tolerated in human participants, with no undesirable physiological effects on markers of liver and kidney function, complete blood counts, hemodynamic parameters, or vital signs [23]. Because this organism is widely distributed among humans and because BCFAs are in foods—notably, dairy products—this treatment is expected to be safe [24].

Although work is still emerging, animal models suggest that BCFAs can attenuate inflammation and related GI tract diseases [25]. Studies in rats have shown that a diet containing a mixture of BCFAs (*iso*-14:0, *anteiso*-15:0, *iso*-16:0, *anteiso*-17:0, *iso*-18:0, and *iso*-20:0) reduces necrotising enterocolitis and inflammation [26]. Taormina et al. [25] noted that increased serum *iso*-BCFAs (i.e., *iso*-15:0, *iso*-16:0, *iso*-17:0, and *anteiso*-15:0) in humans is associated with reduced serum C-reactive protein, suggesting lower inflammation markers.

We recently screened numerous *B. subtilis* strains cultured at lab scale and identified a particularly fast-growing strain with high levels of BCFAs (unpublished data), designated “BG01-4™.” The investigational product or the intervention that is planned to be used in the trial is a proprietary registered non-genetically modified, spore-forming, deactivated strain of the *B. subtilis* BG01-4™ organism. The *B. subtilis* genome was found to have close clusters to *B. subtilis* natto VK161, sharing 99.73% (unpublished), and is also a generally recognised as safe (GRAS)-approved probiotic strain. BG01-4™ is derived by fermentation, deactivated, and freeze dried in a good manufacturing practice (GMP) facility. We prepared spores (5 × 10^9^ colony-forming units) of deactivated *B. subtilis* BG01-4^TM^ at 85 °C for 60 min in a maltodextrin carrier after heat deactivation of vegetative cells into a powdered form in sachets that can be mixed into drinks or foods and consumed as a probiotic.

Clinical and interventional *B. subtilis* studies have previously been shown to control intestinal hydrogen production [27]. Supplementation has also been reported to promote improvements in those with constipation and diarrhoea [4]. A recent clinical trial found that a formulation containing *B. subtilis* significantly reduced the frequency and severity of abdominal bloating [28]. However, well-designed placebo-controlled randomised control trials (RCTs) investigating *B. subtilis* efficacy on overall quality of life in people with FGID have not yet been undertaken.

The aim of this double-blinded RCT was to test the efficacy of a proprietary strain of deactivated *B. subtilis* (BG01-4™) in sachet form to improve symptoms of participants with self-reported FGID. We hypothesised that those who receive 5 × 10^9^ colony-forming units of probiotic BG01-4™ per day over four weeks would improve self-reported symptoms associated with FGID via the Gastrointestinal Symptom Rating Scale (GSRS) [29].

## 2. Materials and Methods

This study was approved by Bellberry Human Research Ethics Committee Australia (2021-09-1137), conformed to the CONSORT 2010 guidelines, and was registered as a clinical trial with the Australian and New Zealand Clinical Trials Registry (ACTRN12621001441808p). All study procedures were conducted in accordance with the ethical principles in the Declaration of Helsinki. Prior to participation, the study was adequately explained, and voluntary written and informed consent to participate in the study was obtained prior to any data collection.

### 2.1. Participants

Healthy adults (n = 67) between 18 and 75 years of age (all: 49.9 ± 14.1 years; males: n = 25 53.5 ± 12.9 years; females n = 42 48.8 ± 14.2 years) were recruited via word of mouth and social media announcements. Functional gastrointestinal disorders often exist in people with no attribution to formal medical diagnosis; therefore, inclusion for participation in the study involved meeting modified Rome IV (https://theromefoundation.org/rome-iv/rome-iv-criteria/ accessed 6 September 2021) criteria [30], which included a self-diagnosis of FGID with symptom onset that must have begun at least 6 months prior and with the criteria having been met for at least 3 months, and having experienced one or more of the following self-reported symptoms for 6 months or longer: (1) recurrent abdominal bloating and/or distension occurring on average at least once per day/week and abdominal bloating and/or distension predominating over other symptoms, such as minor bowel movement abnormalities; (2) loose or watery stools, without predominant abdominal pain or bothersome bloating, occurring in more than 25% of stools; and (3) functional constipation: straining during more than 25% of defecation per week, a sensation of anorectal obstruction/blockage or incomplete evacuation during more than 25% of defecations, needing manual manoeuvres (e.g., digital assistance) to help with more than 25% of defecations, and loose stools being rare without the use of laxatives. Exclusion criteria included current use of probiotics, intolerance to maltodextrin, meeting the criteria for opioid-induced constipation (OIC), or diagnoses of irritable bowel syndrome (IBS), including diarrhea-predominant IBS (IBS-D). Sample size determination was based on a priori power calculation for the primary outcome variables (total score of GSRS) pre- and post-intervention (3 time points) for a between-groups (experimental vs. placebo) repeated measures design suggesting an effect size *f* = 0.2, power (*1-beta*) = 0.95, and α ≥ 0.05 for a total minimum sample size of 66.

### 2.2. Study Design

We used a randomised, double-blind, placebo-controlled, two-arm parallel group study of a four-week supplementation intervention period. Prior to intervention, participants who met the inclusion criteria were provided with a password-protected online link to complete a modified GSRS survey (mGSRS) [29]. Following completion of the mGSRS, participants were randomly chosen to receive BG01-4™ or a placebo (maltodextrin) once daily for four weeks. Follow-up surveys were completed by participants at the mid-point (two-week) mark and following completion of the intervention after four weeks. Individuals were reminded to maintain their current physical activity levels and to continue on their usual diet throughout the study. Participants were also instructed to record any adverse events and to contact researchers if they stopped taking the supplement.

A simple randomisation schedule was manually prepared by an unblinded person who facilitated pre-screening and written informed consent. Participants who met the inclusion criteria were assigned a unique code number based on the randomisation schedule generated by the unblinded research member who was not involved in conducting the study assessment.

### 2.3. Intervention

The interventional supplements (5 × 10^9^ colony-forming units of probiotic BG01-4™ per day over four weeks and placebo) provided by Vernx Pty Ltd. (Melbourne, Victoria, Australia) via mail comprised 1 g unmarked sachets with the instruction to consume one sachet per day for four weeks. Participants were instructed that they could either pour the contents of the sachet into a glass of water (250 mL) or juice or sprinkle it over their usual foods, such as yogurt or cereal. Experimental and placebo treatments were indistinguishable by visual inspection.

### 2.4. Outcome Measures

Three primary outcomes were pre-planned: Total GSRS score, GSRS-constipation, and GSRS-diarrhoea scores were compared for change from entry at four weeks. Pre-planned secondary outcomes were indigestion, dyspepsia, and abdominal pain syndrome, which were similarly tested for change from entry at four weeks.

The GSRS questionnaire is well documented and has a good reliability, with a Cronbach’s alpha ranging from 0.79 to 0.83 across studies across a number of different languages [31,32,33,34]. We used a modified version of the original GSRS (mGSRS) containing 15 questions that were previously validated [34,35] and self-administered online [36,37,38] that evaluated the severity of five symptom clusters: abdominal pain, dyspepsia syndrome, indigestion, constipation, and diarrhea. Participants answered questions using a Likert scale from one (least severe) to seven (most severe) [5,37].

### 2.5. Statistical Analysis

#### 2.5.1. Primary Outcomes

Analyses were performed using intention-to-treat analysis. Data were found to be normally distributed. The difference between scores at four weeks and at entry were tabulated and tested by one-way *t*-test for non-zero, with significance (α ≤ 0.05) adjusted for multiple (three) comparisons (*p* = 0.0167 = 0.05/3). Differences in improvement between the experimental and placebo treatments were tested by a pairwise t-test similarly adjusted for multiple comparisons.

#### 2.5.2. Exploratory Outcomes

For exploratory analysis, responses (pre-, mid-, post-intervention) were combined within the five cluster areas, plus the total score of the mGSRS, averaged between groups. The mean change scores from pre-intervention to mid-intervention and from mid-intervention to post-intervention were calculated and analysed using a mixed-model ANOVA, except for age, for which an independent samples *t*-test was used. Where ANOVA was found to be significant, post hoc testing with the Bonferroni correction was employed. All data were analysed using Jamovi [39] and significance was set at α ≤ 0.05.

## 3. Results

A total of 186 individuals were screened, with 80 eligible participants enrolled and randomised in the study (Figure 1). Groups were matched for age and sex (BG01-4^TM^: M = 23/F19, mean age 47.8 ± 14.8 years; placebo: M = 23/F19, mean age 51.9 ± 13.4 years). Compliance in the BG01-4™ group was 81%, with two participants (5%) dropping out due to not adhering to the protocol and six participants (14%) dropping out voluntarily. Similar compliance in the placebo group was found (87%), with one participant leaving due to protocol violation (3%) and four participants (10%) leaving voluntarily (Figure 1).

### 3.1. Primary and Secondary Outcomes

The results of the primary outcomes show that all experimental groups demonstrated an improvement in scores at four weeks, whereas the placebo groups improved for diarrhoea (Figure 2). Constipation scores improved by 33% over the four-week period, whereas the placebo group’s improvement (15%) did not reach significance. Among secondary outcomes, all treatment groups, including the placebo, showed improved over the four weeks. Indigestion was improved compared to that of the placebo but not statistically significant (*p* = 0.079).

### 3.2. Exploratory Outcomes

Scores from the mGSRS across the five clusters are presented in Table 1. Significant interaction effects were observed between time and treatment for dyspepsia (F_1,53_ = 4.10, *p* = 0.04), indigestion (F_1,53_ = 4.32, *p* = 0.04), constipation (F_1,53_ = 12.58, *p* < 0.001), and GSRS total score (F_1,53_ = 10.40, *p* = 0.002). Post hoc analysis revealed group differences at the end of the intervention for dyspepsia (t_53_ = −3.28, *p* = 0.002), indigestion (t_58_ = −2.36, *p* = 0.02), constipation (t_58_ = −3.46, *p* = 0.001), and GSRS total score (t_58_ = −3.331, *p* = 0.002). Only main effects were observed for abdominal pain (F_1,53_ = 7.98, *p* = 0.007).

## 4. Discussion

The investigation of individual bacterial strains to alleviate FGID in people with chronic symptoms is an emerging area of research. However, the safety of *B. subtilis* is well established. Moreover, a number of benefits have been reported for *B. subtilis* in poultry diets, including immune system improvements, nutrient digestion benefits, and general improvements in gut health, immunity, and growth outcomes in animal studies [40,41,42]. An example of fermented food containing *B. subtilis* is natto, a fermented cooked soybean using *B. subtilis* as a bacterial starter, e.g., *B. subtilis* natto VK161. Homma et al. [43] reported that one gram of natto contains 1 × 10^9^ colony-forming units (CFUs) *B. subtilis*. At a typical serving of 175 g of natto per day, one would consume 1.75 × 10^11^ CFUs of *B. subtilis* per day. *B. subtilis* is also found naturally in human breast milk as healthful probiotic bacteria that support healthy microflora in infants [44]. *B. subtilis* cells are rod-shaped, Gram-positive bacteria. The abundance of these bacteria in the environment and the constant exposure of mammalian species to them support the natural tolerance for *B. subtilis* and reflect their general safety.

*B. subtilis* as a probiotic has been shown to influence several aspects of the human gut, including motility, epithelial strength, and inflammation [45], promoting improvements in those with constipation and diarrhoea [4]. A clinical study found that a formulation containing *B. subtilis* could significantly reduce the frequency and severity of abdominal bloating and pain by 33% and 41%, respectively [28].

In this study, we show that constipation improved in four weeks when a proprietary strain of *B. subtilis* (BG01-4^TM^) with 5 × 10^9^ colony-forming units was consumed every day over four weeks versus the placebo, which was one of three registered primary outcomes. Exploratory analysis showed improvements in participants’ responses for dyspepsia and indigestion.

The placebo used in this study (maltodextrin) was indistinguishable from the *B subtilis* treatment. An interesting finding from the exploratory aspect of our study reflected the GSRS findings by Shi et al. [46] and Östlund-Lagerström et al. [47]. In the study by Shi et al. [46], over the course of a four-week trial, the first two weeks showed self-reported improvements in both groups (~40%) before the placebo group deteriorated in self-reported symptoms (16%). Östlund-Lagerström et al. [47] showed similar outcomes at follow-up 1 (change score −0.04 ± 0.6 vs. −0.08 ± 0.6, respectively) and at follow-up 2 (−0.07 ± 0.6 vs. −0.08 ± 0.6, respectively) between the probiotic and the placebo. In our study, both groups improved in the first two weeks, possibly due to expectation bias. However, the separation at four weeks, where the placebo group reported a worsening of previously improving symptoms (at two weeks) compared to the BG01-4^TM^ arm, suggests a physiological response. Expectation bias is a concern, as with rapid improvements, individuals may discontinue supplementation therapy and potentially return to restrictive dieting [10]. As currently there is no cure for FGID, it is important to manage psychological aspects of this condition through continued supplementation and using a holistic approach via the biopsychosocial model [48].

We observed significant improvements for items within the GSRS. For example, our data for self-reported symptoms for functional dyspepsia showed similar improvements (20%) to a recent exploratory study demonstrating the efficacy of *B. subtilis* MY02, which reported a 20–30% improvement, which is higher than the suggested clinically meaningful outcome of 10–15% [49,50]. Similarly, indigestion symptoms were improved, supporting previous work showing short-term decreases (up to 8 weeks) in anxiety associated with indigestion and abdominal pain [47], as well as relief from constipation, with Labellarte and colleagues reporting increased bowel movements [51]. Taken together, these findings suggest that *B. subtilis* has the potential to significantly relieve FGID symptoms. However, although the exact mechanism of action is unknown, we would propose the following hypothesis: Apart from shifting the distribution of microbiota in the lower gut, *B. subtilis* produces abundant BCFAs that have numerous putative effects in the GI tract. BCFAs are rapidly taken up into enterocytes, where they are incorporated into membrane lipids and modulate the inflammatory response to lipopolysaccharide challenge [12,14]. They are effective in reducing the incidence of necrotising enterocolitis in a neonatal rat model, where they also attenuate the inflammatory response [26]. Our current data are consistent with BCFAs as a contributor to the observed efficacy of *B. subtilis*, and we aim to test this hypothesis in future research.

Overall, the supplement was well tolerated, with the number of dropouts (34%) being slightly higher than that of previous *B. subtilis* research (Figure 1)—for example, that of Wauters et al. (25%) [49]. There was one adverse event reported in the placebo group, but the participant was assessed medically and the event was deemed unlikely to be related to the study product, as the participant had not taken the sachet for a day prior to the incident.

Although we measured mGSRS as a proxy for overall quality of life, which showed general psychometric improvements, we did not specifically measure mood or depression/anxiety, which is a limitation of the study. However, it was outside the scope of this study to quantify psychological state, as the research aim was to address individuals with chronic FGID who would otherwise be psychologically healthy. We do acknowledge, however, that future RCTs should include those with FGID who have clinically diagnosed mood disorders resulting from chronic gastrointestinal disorders.

Although in this phase 1/2a trial we found statistically significant differences between groups, the duration of treatment was brief in our study (four weeks), which has also been acknowledged in previous work, including those of Penet et al. and Hanifi et al. [5], who also ran four-week trials with limited results. Although some probiotic studies report that an intervention of at least 60 days is useful for statistical significance [52], others have suggested this period is not long enough to see the full effects on gastrointestinal disorders [28]. Based on the data in this study, we aim to complete extended trials in the future to comply with the longer treatment periods required.

A further limitation of this study was that we were not in a position to control for diet or prebiotic use, as our research question involved investigating the efficacy of adding probiotics to an individual’s current diet. We were also not able to control for individuals consuming fermented food in their current diet. We acknowledge that those consuming fermented food may have diluted our findings.

In conclusion, consumption of a proprietary strain of probiotic *B. subtilis* (BG01-4^TM^) was well tolerated and resulted in relief of symptoms of constipation and related GI dysfunction in this randomised double-blinded placebo-controlled trial. Although longer studies are required, statistically significant improvements in overall GSRS suggest that BG01-4^TM^ has a potential role in relieving specific symptoms of dyspepsia, indigestion, and constipation after four weeks.

## Figures and Tables

**Figure 1 nutrients-15-04490-f001:**
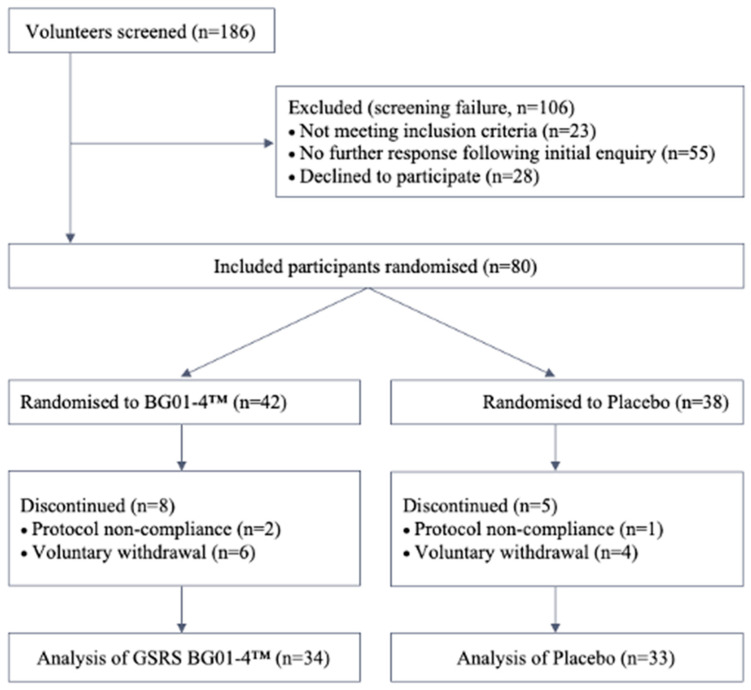
Study participant screening and selection.

**Figure 2 nutrients-15-04490-f002:**
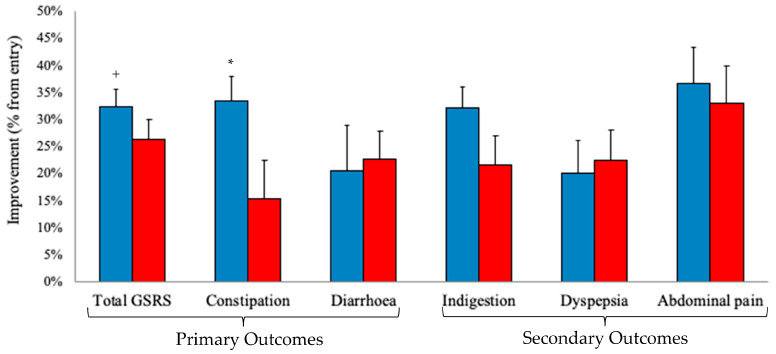
Improvement in reported GI symptoms (mean ± SD). Blue columns represent the BG-1-4^TM^ group. Red columns represent the placebo group. Primary outcomes: total GSRS, constipation, and diarrhoea. Constipation improved for the experimental BG01-4^TM^ group but not the control, with the alpha level (0.05) adjusted for multiple comparisons (* *p* = 0.0167 = 0.05/3). Diarrhoea was significantly improved for both the experimental and placebo groups. The total GSRS score improved by 11% compared to the placebo (^+^
*p* < 0.05 improvement over 4 weeks). Pre-planned secondary outcomes: All experimental and placebo groups were improved. Indigestion improved in the BG01-4^TM^ group by 32% compared to the placebo group (21%; *p* = 0.079). Constipation and indigestion are both primarily controlled by lower GI tract function.

**Table 1 nutrients-15-04490-t001:** Mean (±SD) mGSRS scores between groups across the five symptom clusters (* *p* > 0.05 compared to baseline).

Group	Abdominal Pain			Dyspepsia			Indigestion			Constipation			Diarrhoea			Total Score		
	Pre	2-wk	4-wk	Pre	2-wk	4-wk	Pre	2-wk	4-wk	Pre	2-wk	4-wk	Pre	2-wk	4-wk	Pre	2-wk	4-wk
BG01-4™	7.22 (±2.01)	5.15 (±2.25)	4.41 (±2.15)	6.26 (±2.65)	5.41 (±2.66)	4.81 * (±2.57)	15.00 (±3.55)	11.30 (±3.61)	10.00 * (±3.41)	11.67 (±4.88)	9.93 (±4.58)	7.85 * (±4.46)	8.41 (±4.28)	7.44 (±4.26)	5.81 (±2.87)	48.56 (±9.45)	39.22 (±10.56)	32.89 * (±10.60)
Placebo	7.43 (±1.89)	5.07 (±2.68)	4.86 (±2.52)	8.36 (±4.00)	6.11 (±3.76)	6.29 (±3.46)	15.29 (±4.38)	10.71 (±5.16)	11.14 (±4.58)	10.43 (±4.52)	7.55 (±4.13)	8.32 (±4.08)	10.79 (±5.45)	7.39 (±4.23)	8.18 (±5.13)	52.64 (±10.89)	36.96 (±14.67)	39.07 (±14.04)

## Data Availability

The datasets used and/or analysed during the current study are available from the corresponding author on reasonable request.

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
