# Peer review of "Bacillus Subtilis (BG01-4TM) Improves Self-Reported Symptoms for Constipation, Indigestion, and Dyspepsia: A Phase 1/2A Randomized Controlled Trial"

_nutrients, 2023, doi:10.3390/nu15214490_

Round 1

Reviewer 1 Report

The manuscript briefly described interesting findings from a double blinded randomized controlled trial of deactivated Bacillus subtilis (BG01-4™). The results suggested that in people with self-reported FGID, the strain BG01-4™ improved symptoms of constipation, GI dysfunction, potentially mediated via synthesis of BCFA. 

Several minor corrections and improvements should be done before its publication.  

Dear Authors, 

- Please Introduce in the Title the complete and correct name of the strain Bacillus subtilis  (BG01-4™)- italics for the species. 

- Please reformule the Title as it is not demonstrated that BCFA is linked to beneficial effects e.g. "Bacillus subtilis (BG01-4™) improves self-reported symptoms for constipation, indigestion and dyspepsia. A phase 1/2A randomized controlled trial"

- Please describe better and support with experimental data features of the strain BG01-4™ regarding absence of safety concerns (GRAS, QPS, etc), fast-growing strain with high levels of BCFA, especially the BCFA production as it is key factor mediating the potential beneficial effects. 

- Please describe better the preparation of the spores of deactivated B. subtilis BG01-4TM (temperature, active colonies, initial/final CFU or AFU, etc). 

- The following statement does not fully apply for presenting the aim of the study: 

"However, well designed placebo-controlled randomized control trials (RCTs) investigating B subtilis efficacy on mental health and overall quality of life in people with FGID have not yet been undertaken". 

Please delete it or rephrase accordingly. 

 Please provide data or Annex for the modified GSRS survey (mGSRS), is it appropriatly validated?

Figures sould be of better quality (pixels).

Figure 2 should be better shown in one graphic a+b, which will allow to see all the parameters together.

Table 1: Abdominal should tell explicitely Abdominal pain or discomfort?

p values should be indicated in the Table corresponding values with * or similar symbols

At the discussion, please introduce more scientific evidence data regarding the beneficial effects and uses for which this specific strain is already commercialised. 

Author Response

The manuscript briefly described interesting findings from a double blinded randomized controlled trial of deactivated Bacillus subtilis (BG01-4™). The results suggested that in people with self-reported FGID, the strain BG01-4™ improved symptoms of constipation, GI dysfunction, potentially mediated via synthesis of BCFA. 

Several minor corrections and improvements should be done before its publication.  

Dear Authors, 

- Please Introduce in the Title the complete and correct name of the strain Bacillus subtilis  (BG01-4™)- italics for the species. 

- Please reformule the Title as it is not demonstrated that BCFA is linked to beneficial effects e.g. "Bacillus subtilis (BG01-4™) improves self-reported symptoms for constipation, indigestion and dyspepsia. A phase 1/2A randomized controlled trial"

Response: We have retitled the paper and have included the strain name as well.

- Please describe better and support with experimental data features of the strain BG01-4™ regarding absence of safety concerns (GRAS, QPS, etc), fast-growing strain with high levels of BCFA, especially the BCFA production as it is key factor mediating the potential beneficial effects. 

Response: We have added wording in lines 111-118 that should address the Reviewer’s comment here. We had to limit some details as they are proprietary and commercial in confidence. We hope the Reviewer appreciates this.

- Please describe better the preparation of the spores of deactivated B. subtilis BG01-4TM (temperature, active colonies, initial/final CFU or AFU, etc). 

Response: We have added wording in lines 118-121 to better explain the preparation of spores.

- The following statement does not fully apply for presenting the aim of the study: 

"However, well designed placebo-controlled randomized control trials (RCTs) investigating B subtilis efficacy on mental health and overall quality of life in people with FGID have not yet been undertaken". 

Please delete it or rephrase accordingly. 

Response: We agree with the Reviewer that this RCT isn’t specifically measuring mental health and have removed the wording to reduce confusion (now line 126).

Please provide data or Annex for the modified GSRS survey (mGSRS), is it appropriatly validated?

Response: We have added citations now specifically regarding the modified GSRS (line 201).

Figures sould be of better quality (pixels).

Response: We will submit to the editorial office TIFF version of the figures for better quality which may have been reduced in the submission process.

Figure 2 should be better shown in one graphic a+b, which will allow to see all the parameters together.

Response: While we appreciate the suggestion from Reviewer 1, the reason we split the figure was that there were primary and secondary outcomes and aimed to be consistent with our clinical trial registration. We have combined this into one graphic (figure 2) and have added “primary” and “secondary” to not confuse readers. (Please excuse the quality, we will send a high-resolution version to the Editorial Office)

Table 1: Abdominal should tell explicitely Abdominal pain or discomfort?

Response: We have added “pain” in Table 1 and thank the Reviewer for this observation

p values should be indicated in the Table corresponding values with * or similar symbols

Response: We have now added the asterisk to indicate significance.

At the discussion, please introduce more scientific evidence data regarding the beneficial effects and uses for which this specific strain is already commercialised.

Response: Extra wording (lines 274-291) to introduce the background scientific evidence regarding the beneficial effects and uses of B Subtilis has been added.

Reviewer 2 Report

The authors present an interesting study, in human volunteers, exploring the effect of a strain of Bacillus subtilis on functional gastroenteritis disorders. The results are promising, but their robustness is not shown (conclusions are correct?), as the Discussion section must be greatly improved.

Specifically minor and major comments are:

Fig. 2b: Are the height of the Dyspepsia columns correct?

Discussion Section: The authors must improve the Discussion section to support the strength of its results and conclusions. References from number 38 onwards are missing in the References Section, which makes the work of revision difficult. Why haven't the results of references 5 or 20 been commented on?. The authors should also compare the effect of the tested strain with the effect of other microorganisms used as probiotics for FGIDs. Bacillus subtilis strains are often used as probiotic in production animals, the authors could discuss whether any similar effect or not has been found in animals. The authors could compare the results obtained in this study with other studies where BCFA are directly ingested. Please try make comparisons with updated references as most of the references cited are old.

The authors should better explain the findings of other works they use to support their study, as they should not force the reader to do that work, examples, line 260: “showed similar improvements to a recent exploratory study demonstrating the efficacy of B subtilis MY02 versus placebo”….explain more specifically what are those “similar” results. Line 262 Refs 39 and 42: how much “improved”?. Do the same for the other references to be used.

References: missing references from number 38 onwards

Ref. 5: pages?, italics for the microorganism (also ref. 17)

Ref 7: lack of dots

Author Response

The authors present an interesting study, in human volunteers, exploring the effect of a strain of Bacillus subtilis on functional gastroenteritis disorders. The results are promising, but their robustness is not shown (conclusions are correct?), as the Discussion section must be greatly improved.

Specifically minor and major comments are:

Fig. 2b: Are the height of the Dyspepsia columns correct?

Response: We have rechecked the data and the Dyspepsia columns are correct.

Discussion Section: The authors must improve the Discussion section to support the strength of its results and conclusions. References from number 38 onwards are missing in the References Section, which makes the work of revision difficult. Why haven't the results of references 5 or 20 been commented on?. The authors should also compare the effect of the tested strain with the effect of other microorganisms used as probiotics for FGIDs. Bacillus subtilis strains are often used as probiotic in production animals, the authors could discuss whether any similar effect or not has been found in animals. The authors could compare the results obtained in this study with other studies where BCFA are directly ingested. Please try make comparisons with updated references as most of the references cited are old.

Response: We thank the Reviewer for bringing to our attention that a number of citations had not been included in the end-text reference list. We are unsure why a number of references were dropped off in the manuscript submitted, given we used EndNote, but this has now been rectified. We have also added discussion on similar benefits in animal studies (lines 274-278) We have also expanded wording in lines 329-335 to include comments regarding studies by Penet et al and Hanifi et al.

The authors should better explain the findings of other works they use to support their study, as they should not force the reader to do that work, examples, line 260: “showed similar improvements to a recent exploratory study demonstrating the efficacy of B subtilis MY02 versus placebo”….explain more specifically what are those “similar” results. Line 262 Refs 39 and 42: how much “improved”?. Do the same for the other references to be used.

Response: We appreciate the Reviewer’s concern and have added wording to expand our findings with the literature cited (lines 309-316).

References: missing references from number 38 onwards

Response: We have added the missing references. We apologize for this but can only assume something happened in the submission process.

Ref. 5: pages?, italics for the microorganism (also ref. 17)

Response: We thank the Reviewer for noticing this. We have added page numbers and italics for microorganism for reference 5 and also for reference 17.

Ref 7: lack of dots

Response: Dots now added.

Reviewer 3 Report

The authors said that B. subtilis capable of producing BCFA can improve symptoms for constipation, indigestion and dyspepsia. However, I do not see a correlation between the metabolites and what the authors will prove. In general, how do you say that it is the produced bcfa to have the effects reported? There is no biochemical data detecting the BCFAs or analyses of the intestinal microbial community that support this claim.

Abstract:

add quantitative data, especially in the part of the results (lines 21-27), when you said for example “At four weeks one of three primary outcomes, constipation in the experimental group, was improved compared to placebo”

Introduction:

In general better elucidate the role of BCFAs in the pathological condition described and motivate well the choice of that Bacillus strain.

Line 37-38: characterized ”by”

Methods:

Lines 125-126: “Healthy adults between 18-75 years of age (all: 49.9 ± 14.1 years; males: n=25 53.5 ± 125 12.9 years; females 48.8 ± 14.2 years”. You have to add the number of females and correct the grammar.

Lines 140-147: do you ask about the diet? Subject can eat also fermented food? Sometimes Bacillus is included in the preparation of some cheese or other foods. Do you ask about the consumption of possible prebiotic molecules? If possible, include in the text some more details about inclusion-exclusion criteria, to help to understand the results obtained

Lines 149-163: how do you conduct the randomization? Manually or did you use a software? Please specify

Results:

Figure 2a and 2b: please add the statistic symbols (es * for p<0.05) and improve figure quality

In general, keep attention to the punctuation, the correct spelling, and correct the typing errors. The text is understandable and the results well presented.

Author Response

The authors said that B. subtilis capable of producing BCFA can improve symptoms for constipation, indigestion and dyspepsia. However, I do not see a correlation between the metabolites and what the authors will prove. In general, how do you say that it is the produced bcfa to have the effects reported? There is no biochemical data detecting the BCFAs or analyses of the intestinal microbial community that support this claim.

Response: We appreciate the Reviewer’s concern, however we have only discussed BCFA’s in the Introduction as providing the rationale for the study. We haven’t discussed this as a mechanism for our findings in the discussion. We have, however, removed the wording “…potentially mediated via synthesis of BCFA” in the abstract (line 29). We have also removed the wording “Branched fat synthesizing…” in the title as per the request of the other Reviewer.

Abstract:

add quantitative data, especially in the part of the results (lines 21-27), when you said for example “At four weeks one of three primary outcomes, constipation in the experimental group, was improved compared to placebo”

Response: Quantitative data has been added (lines 22-25).

Introduction:

In general better elucidate the role of BCFAs in the pathological condition described and motivate well the choice of that Bacillus strain.

Line 37-38: characterized ”by”

Response: We have added wording in lines 104-109 to illustrate the role of BCFAs in disease further to describing BCFAs in health.

Methods:

Lines 125-126: “Healthy adults between 18-75 years of age (all: 49.9 ± 14.1 years; males: n=25 53.5 ± 125 12.9 years; females 48.8 ± 14.2 years”. You have to add the number of females and correct the grammar.

Response: We apologize for this oversight and have fixed up the typographical errors with this sentence and data (lines 143-144).

Lines 140-147: do you ask about the diet? Subject can eat also fermented food?

Sometimes Bacillus is included in the preparation of some cheese or other foods. Do you ask about the consumption of possible prebiotic molecules? If possible, include in the text some more details about inclusion-exclusion criteria, to help to understand the results obtained

Response: Our study controlled for only those not having probiotics in their diet. We were not able to control for anything beyond supplementary probiotics as our research question was to investigate the efficacy of probiotics on top of the individual’s current diet. We have acknowledged this in the limitations section of the Discussion (lines 338-342). We have also acknowledged in the same section that those eating fermented food may dilute our results.

Lines 149-163: how do you conduct the randomization? Manually or did you use a software? Please specify

Response: We have added the word manually. The sentence now reads “A simple randomization schedule was manually prepared by an unblinded person who facilitated pre-screening and written informed consent (see lines 180).

Results:

Figure 2a and 2b: please add the statistic symbols (es * for p<0.05) and improve figure quality

Response: We have added the asterisk and a plus symbol to correspond to the figure legend. As per the other Reviewer’s request we have now collapsed the two figures into one.  We will submit to the editorial office TIFF versions of the figure for better quality which may have been reduced in the submission process.

In general, keep attention to the punctuation, the correct spelling, and correct the typing errors. The text is understandable and the results well presented.

Response: We thank the Reviewer for their feedback and appreciate the typographical errors picked up.

Round 2

Reviewer 1 Report

The manuscript has been revised according to main suggestions. However, the beneficial effects of the strain cannot be associated to a specific mechanism, as initially was proposed to branched fat.

Author Response

The manuscript has been revised according to main suggestions. However, the beneficial effects of the strain cannot be associated to a specific mechanism, as initially was proposed to branched fat.

Response: This is a good point raised by the Reviewer and we agree. We have move wording from orginally lines 92-99 (sentences starting with “Apart from shifting the distribution…”) and moved this to the discussion section in lines 317-326 to propose a hypothesis, rather than causation/association, to allow for further research. Further, we have changed specific wording in the abstract that implied an association between BCFAs with a specific mechanism as highlighted by the Reviewer (lines 15 and 17).

Reviewer 2 Report

The authors have made a good effort, and the manuscript is much improved. However, there are still some minor questions to be answered.

I find differences between the data indicated in Table 1 and Fig 2a (i.e. diarrhea: wouldn't the value be 30,9 % for blue?.. or I am wrong?). If there are discrepancies, does this affect the exploratory outcomes paragraph?.

Fig.2.a. : the explanation of the colors does not appear in the new figure.

Lines 295-299: the similarities between the results obtained in this study and the results of reference 49 are still unclear, please add numerical data (i.e.: similar, significant as well…are quite unconcreted assertions).

Discussion section, authors could consider this indication: human trials are very important and rare, so this is one of the strengths of this work. And for this same reason, if the authors make comparisons of their results with other researchers (examples: 49, 50, etc.) they should indicate some numerical data from the other researchers tests, such as number of individuals, time, significance level (number), from the other trials, etc., and thus in this way the reader of this manuscript can confirm general assertions beyond the: they match our work.

Author Response

The authors have made a good effort, and the manuscript is much improved. However, there are still some minor questions to be answered.

I find differences between the data indicated in Table 1 and Fig 2a (i.e. diarrhea: wouldn't the value be 30,9 % for blue?.. or I am wrong?). If there are discrepancies, does this affect the exploratory outcomes paragraph?.

Response: We appreciate the Reviewer’s question and have rechecked the data. The appearance of difference is that in Figure 2, the data is expressed as a mean percentage change from pre to post, which includes both positive and negative percentage changes, whereas the raw data scores (all positive numbers) from the mGSRS is tabulated. So the grouped difference between the raw scores in Table 1 does not take into account the percentage differences in all subjects as expressed in Figure 2. We see this discrepancy in other items, for example Indigestion has a 32% and 21% improvement for BG01-4 and placebo respectively in Figure 2, but if you look at the raw difference it comes to 33.3% and 27.1% respectively. We hope that this explains the apparent discrepancy.

Fig.2.a. : the explanation of the colors does not appear in the new figure.

Response: We thank the Reviewer for noticing this and apologies for this oversight. We have also removed the “a” from figure 2a as there is once combined figure now.

Lines 295-299: the similarities between the results obtained in this study and the results of reference 49 are still unclear, please add numerical data (i.e.: similar, significant as well…are quite unconcreted assertions).

Response: We have reworded sentence as the Wauters et al citation [#49] as it was referring to the percentage of responders which we did not measure. But we have now included the numerical change in both this study and the Wauters et al paper (lines 326-327).

Discussion section, authors could consider this indication: human trials are very important and rare, so this is one of the strengths of this work. And for this same reason, if the authors make comparisons of their results with other researchers (examples: 49, 50, etc.) they should indicate some numerical data from the other researchers tests, such as number of individuals, time, significance level (number), from the other trials, etc., and thus in this way the reader of this manuscript can confirm general assertions beyond the: they match our work.

Response: We appreciate the reviewer’s comment here and have included more specific data throughout the Discussion section to illustrate matching of the work.

Reviewer 3 Report

The authors improved the text with the suggestions. Although for me the study is not so innovative, it is written correctly and supported by good results. 

Author Response

The authors improved the text with the suggestions. Although for me the study is not so innovative, it is written correctly and supported by good results. 

Response: We thank the Reviewer for their comments